# Implication of 5-HT in the Dysregulation of Chloride Homeostasis in Prenatal Spinal Motoneurons from the G93A Mouse Model of Amyotrophic Lateral Sclerosis

**DOI:** 10.3390/ijms21031107

**Published:** 2020-02-07

**Authors:** Elodie Martin, William Cazenave, Anne-Emilie Allain, Daniel Cattaert, Pascal Branchereau

**Affiliations:** University of Bordeaux, CNRS, INCIA, UMR 5287, F-33000 Bordeaux, France; elo.martin33@gmail.com (E.M.); william.cazenave@u-bordeaux.fr (W.C.); anne-emilie.allain@u-bordeaux.fr (A.-E.A.); daniel.cattaert@u-bordeaux.fr (D.C.)

**Keywords:** ALS, development, SOD1^G93A^ mouse, 5-HT, chloride homeostasis, GABA/glycine, perforated patch-clamp, spinal cord, motoneuron

## Abstract

Amyotrophic lateral sclerosis (ALS) is a fatal neurodegenerative disease characterized by progressive motor neuron degeneration and muscle paralysis. The early presymptomatic onset of abnormal processes is indicative of cumulative defects that ultimately lead to a late manifestation of clinical symptoms. It remains of paramount importance to identify the primary defects that underlie this condition and to determine how these deficits lead to a cycle of deterioration. We recently demonstrated that prenatal E17.5 lumbar spinal motoneurons (MNs) from SOD1^G93A^ mice exhibit a KCC2-related alteration in chloride homeostasis, i.e., the E_GABAAR_ is more depolarized than in WT littermates. Here, using immunohistochemistry, we found that the SOD1^G93A^ lumbar spinal cord is less enriched with 5-HT descending fibres than the WT lumbar spinal cord. High-performance liquid chromatography confirmed the lower level of the monoamine 5-HT in the SOD1^G93A^ spinal cord compared to the WT spinal cord. Using ex vivo perforated patch-clamp recordings of lumbar MNs coupled with pharmacology, we demonstrated that 5-HT strongly hyperpolarizes the E_GABAAR_ by interacting with KCC2. Therefore, the deregulation of the interplay between 5-HT and KCC2 may explain the alteration in chloride homeostasis detected in prenatal SOD1^G93A^ MNs. In conclusion, 5-HT and KCC2 are two likely key factors in the presymptomatic phase of ALS, particular in familial ALS involving the SOD1^G93A^ mutation.

## 1. Introduction

Amyotrophic lateral sclerosis (ALS), also known as Lou Gehrig’s disease, is a rapidly progressive neurodegenerative disease that targets motor neurons. It is one of the most common and most devastating neurodegenerative diseases. In 90% of cases, ALS is idiopathic and sporadic (sALS). Only 10% of ALS cases are familial in origin (fALS) and are inherited in an autosomal dominant manner. Approximately 20 genes are associated with ALS, with the most common causes of typical ALS being mutations in SOD1, TARDBP, FUS, and C9orf72 [1]. ALS is a lethal adult-onset neurodegenerative disease, and most studies have focused on symptomatic adult stages. However, a growing array of evidence has indicated that the disease involves early abnormal development of motor neurons beginning in embryonic stages [2,3]. The serotonergic system has been shown to be involved in ALS and appears to be disrupted very early in development [4,5,6]. 5-HT can impair the development of motor neurons at early embryonic stages and participate in the course of the disease.

The central neurons that produce the biogenic amine serotonin (5-hydroxytryptamine, 5-HT) form a small population of 20–30,000 neurons in the rat central nervous system (CNS) [7] called the B1–B9 cell groups [8]: a caudal cluster in the medulla corresponds to the raphe pallidus, obscurus and pontis (B1–B5), and a rostral cluster in the pons includes the dorsal raphe and median raphe (B6–B9) [9]. Raphe neurons endowed with profuse collateralizations of their axons provide extensive and diffuse innervation to all brain areas and the spinal cord (SC), with the caudal raphe group projecting to the SC and the rostral group projecting to the forebrain. Raphe 5-HT neurons are among the earliest to be detected in the developing central nervous system [5]. The rostral raphe group appears earlier (at embryonic day (E)10.5) and is induced by the coordinated expression of two morphogens, Fgf8 and Shh, whereas the caudal cluster appears 24 h later and is regulated by the cooperation of Fgf4 and Shh [10]. The development of descending serotonergic pathways from neurons located in the raphe nuclei has been described in the embryonic rat SC, where they are present in the cervicothoracic cord at E14 and reach the lumbar cord at E16–17 [11]. In a previous study, we described the ontogeny of descending 5-HT pathways in the mouse SC [12]. We found that descending 5-HT-immunoreactive (5-HT-ir) axons reach thoracic levels at E13.5 and lumbar levels at E15.5. Some 5-HT-ir fibres are detected in the ventral and intermediate grey matter by E15.5, whereas the dorsal grey matter is not invaded before birth [12].

5-HT neurotransmission is effective early during development [13] and is subject to numerous developmental effects [14]. We have shown that the maturation of the inhibitory system in the mouse embryonic SC is controlled by descending 5-HT raphe inputs [15,16]. This inhibitory system mostly develops after E15.5, when a switch occurs, inducing a substantial drop in the intracellular chloride concentration [Cl^−^]_i_ in lumbar motoneurons (MNs) and rendering GABA unable to provide excitation [17]. This timing corresponds to the arrival of descending 5-HT raphe inputs. The lowering of the [Cl^−^]_i_ during the course of CNS development has been extensively studied, and it has been shown that it mainly relies on the differential ontogenic expression of Na^+^-K^+^-2Cl^−^ cotransporter isoform1 (NKCC1), which takes up chloride ions [18] and neuronal K^+^-Cl^−^ cotransporter type 2 (KCC2), which extrudes chloride ions [19]. KCC2 and NKCC1 are expressed in mouse embryonic lumbar spinal MNs, where they control the maturation of the chloride ion equilibrium (E_Cl_) [17].

Using the SOD1^G93A^ transgenic mouse model (Gly93→Ala substitution in Cu/Zn superoxide dismutase 1, SOD), which expresses a high level of human mutant SOD1 and faithfully recapitulates a vast majority of the pathological abnormalities seen in ALS patients [20], we previously found that SOD1^G93A^ MNs are hyperexcitable prenatally at E17.5 because of shorter dendritic trees and increased input resistance [3]. We also recently showed that chloride homeostasis is altered (i.e., a higher [Cl^−^]_i_) in E17.5 SOD1^G93A^ lumbar spinal MNs because of a specific reduction in KCC2 efficacy [21]. Interestingly, it has been demonstrated that the activation of 5-HT_2AR_ hyperpolarizes the reversal potential of inhibitory postsynaptic potentials in rat spinal MNs through an increase in KCC2 function [22]. In the present report, we show that descending 5-HT fibres are less abundant in the lumbar SC of SOD1^G93A^ mice than in the lumbar SC of WT littermates. We also show that exogenous applications of 5-HT exert potent hyperpolarizing effects on the E_Cl_, assessed here as the reversal potential of GABA_A_R (E_GABAAR_). This leads to the conclusion that the lack of 5-HT innervation in the prenatal SC may be responsible for the alteration in chloride homeostasis in SOD1^G93A^ spinal lumbar MNs and may explain the neurodevelopmental features of ALS that we recently described in the SOD1^G93A^ mouse [3,21].

## 2. Results

### 2.1. 5-HT-ir Descending Fibres in the E17.5 SOD1^G93A^ SC Versus the E17.5 WT SC

Coronal sections of the SC (Figure 1A) revealed that a dense plexus of 5-HT-ir axons was mostly localized in the intermediate area, central grey matter and ventral horn, with the dorsal horn being devoid of 5-HT immunoreactivity. Numerous 5-HT-ir fibres were detected in the lateral funiculi (arrows). Higher magnification (40×) of the cervical SC of WT mice showed numerous axons with varicosities travelling through the white matter of the lateral funiculus and the ventral grey matter of the ventral horn (VH) (Figure 1B1), through the intermediate area (IA) (Figure 1B2) and through the central grey matter (CGM) around the central canal (cc) (Figure 1B3). When quantified in the VH, IA and CGM in coronal sections of the cervical SC, there was no significant difference in 5-HT labelling (*p* > 0.05, Mann-Whitney test) between the WT (VH: 2.21 ± 0.21 AUs (arbitrary units), IA: 3.0 ± 0.25 AUs, CGM: 1.59 ± 0.22 AUs, *n* = 11) and SOD1^G93A^ (VH: 1.72 ± 0.3 AUs, IA: 3.81 ± 0.50 AUs, CGM: 1.29 ± 0.22 AUs, *n* = 11) mice (Figure 1B4–B6). Higher magnification of the coronal sections of the WT lumbar SC revealed the same pattern of 5-HT labelling as that in the cervical sections but a significantly lower density along the cervico-lumbar extent. Interestingly, 5-HT-ir fibres appeared less numerous in SOD1^G93A^ lumbar SC than in the WT lumbar SC (Figure 1C1–C6). This was confirmed by quantitative analysis showing the absence of a significant difference in 5-HT labelling intensity between the SOD1^G93A^ cervical SC and the WT cervical SC (Figure 1D1) and the significant reduction in 5-HT staining in the SC of SOD1^G93A^ mice compared to the SC of WT littermates in the VH (1.86 ± 0.13 AUs and 0.96 ± 0.12 AUs in WT and SOD1^G93A^ SCs, respectively, *n* = 12, *p* < 0.001, Mann-Whitney test), IA (3.31 ± 0.26 AUs and 2.02 ± 0.21 AUs in WT and SOD1^G93A^ SCs, respectively, *n* = 12, *p* < 0.01, Mann-Whitney test), and CGM (1.20 ± 0.14 AUs and 0.48 ± 0.09 AUs in WT and SOD1^G93A^ SCs, respectively, *n* = 12, *p* < 0.001, Mann-Whitney test) (Figure 1D2).

### 2.2. 5-HT Content in the Lumbar SC

To confirm our anatomical results showing a specific reduction in the intensity of 5-HT staining in the lumbar SC, we conducted high-performance liquid chromatography (HPLC) analysis of the spinal content of endogenous 5-HT, a biogenic amine, in the SOD1^G93A^ and WT lumbar SCs. This HPLC analysis revealed that the 5-HT content was significantly reduced in SOD1^G93A^ mice (664.2 ± 53.7 pg/mg, *n* = 24) compared to WT littermates (875.6 ± 84.9 pg/mg, *n* = 16) (*p* < 0.05, Mann-Whitney test) (Figure 2).

### 2.3. Modulation of the E_GABAAR_ by 5-HT

#### 2.3.1. Exogenous 5-HT Hyperpolarizes the E_GABAAR_

In a recent report, we showed that the E_GABAAR_ is more depolarized in E17.5 SOD1^G93A^ lumbar MNs than in the E17.5 lumbar MNs of WT littermates due to KCC2 downregulation [21]. Because 5-HT is able to hyperpolarize the reversal potential of inhibitory postsynaptic potentials in rat spinal MNs by increasing KCC2 function [22], we tested the effect of exogenous 5-HT application on the E_GABAAR_. In MNs voltage-clamped at a holding potential of −70 mV (i.e., close to the physiological resting membrane potential of E17.5 mouse spinal MNs [17]), 5-HT induced a potent inward current, while the E_GABAAR_ became robustly more hyperpolarized (Figure 3A1). Our quantitative analysis performed on E17.5 WT lumbar MNs indicated that the E_GABAAR_ was −56.2 ± 2.9 mV in the control and became significantly more hyperpolarized in the presence of 5-HT (−71.1 ± 2.8 mV, *p* < 0.001, Wilcoxon matched-pairs signed rank test, *n* = 13) (Figure 3A2, left panel). A similar effect of 5-HT was found in E17.5 SOD1^G93A^ lumbar MNs: the E_GABAAR_, which was −47.7 ± 4.2 mV in the control condition (more positive than in WT MNs, *p* < 0.05, Mann-Whitney test), dropped to −67.2 ± 5.2 mV in the presence of 10 µM 5-HT (*p* < 0.001, Wilcoxon matched-pairs signed rank test, *n* = 13) (Figure 3A2, right panel). We also quantified the intensity of the 5-HT-induced inward current and did not find any significant difference between MNs from SOD1^G93A^ mice and those from WT littermates (83.7 ± 10.4 pA (*n* = 18) and 92.3 ± 17.4 pA (*n* = 15) in SOD1^G93A^ and WT mice, respectively, *p* > 0.05, Mann-Whitney test). Finally, we quantified the intensity of E_GABAAR_ hyperpolarization in both genotypes and did not find any difference (−14.9 ± 1.7 mV (*n* = 13) and −19.5 ± 3.1 mV (*n* = 13) in SOD1^G93A^ and WT mice, respectively, *p* > 0.05, Mann-Whitney test).

#### 2.3.2. 5-HT_2R_ Is Involved in E_GABAAR_ Hyperpolarization

As a next step, we verified the specificity of E_GABAAR_ modulation by 5-HT by performing a series of experiments in which 5-HT antagonists that block at least 5-HT_2R_ (10 µM methysergide + 10 µM ketanserine) were applied in addition to exogenous 5-HT (10 µM). This pharmacological protocol revealed that both the inward current and E_GABAAR_ hyperpolarization were reversed when the 5-HTR antagonists were bath-applied in the presence of 5-HT (Figure 3B1). In WT MNs, the E_GABAAR_ dropped from −54.2 ± 6.2 mV to −67.6 ± 3.5 mV in the presence of 5-HT (*p* < 0.05, Wilcoxon matched-pairs signed rank test, *n* = 5) and reversed to −51.4 ± 5.2 mV when the 5-HTR antagonists were applied (Figure 3B2, left panel). In SOD1^G93A^ MNs, the E_GABAAR_ dropped from −54.7 ± 5.7 mV to −71.8 ± 8.3 mV in the presence of 5-HT (*p* < 0.05, Wilcoxon matched-pairs signed rank test, *n* = 8) and reversed to −57.2 ± 6.1 mV when the 5-HTR antagonists were applied (Figure 3B2, right panel).

#### 2.3.3. KCC2 Is Involved in E_GABAAR_ Hyperpolarization

Knowing that the activation of 5-HT_2AR_ hyperpolarizes the reversal potential of inhibitory postsynaptic potentials in rat spinal MNs by increasing the function of KCC2 [22], we further tested the effect of the specific KCC2 blocker VU0240551 on the 5-HT-induced hyperpolarization of E_GABAAR_. We first bath-applied 5-HT and tested its effect on E_GABAAR_. Then, a mixture of VU0240551+ 5-HT was applied, and the E_GABAAR_ was measured. This protocol revealed that blocking KCC2 specifically reversed the 5-HT-induced hyperpolarization of the E_GABAAR_ without affecting the inward current recorded in the MNs (Figure 3C1). In WT MNs, the E_GABAAR_ dropped from −46.7 ± 5.5 mV to −67.1 ± 5.8 mV in the presence of 5-HT (*p* < 0.05, Wilcoxon matched-pairs signed rank test, *n* = 6) and reversed to −47.3 ± 3.4 mV when the KCC2 blocker was added (Figure 3C2, left panel). In SOD1^G93A^ MNs, the E_GABAAR_ dropped from −40.4 ± 4.6 mV to −59.8 ± 7.4 mV in the presence of 5-HT (*p* < 0.05, Wilcoxon matched-pairs signed rank test, *n* = 5) and reversed to −37.0 ± 4.4 mV when VU0240551 was applied (Figure 3C2, right panel).

## 3. Discussion

Our results indicate that the SOD1^G93A^ lumbar SC is less enriched with 5-HT-ir fibres and exhibits a lower 5-HT content than the lumbar SC of WT littermates. These results may reveal a delay in the development of 5-HT-ir descending fibres in SOD1^G93A^ animals. We recently described reduced GABA/glycine synaptic input in E17.5 lumbar SOD1^G93A^ MNs that might reflect a developmental delay in the maturation of SOD1^G93A^ SCs compared to WT SCs. Because the soma perimeter and the surface area of E17.5 WT MNs and SOD1^G93A^ MNs have been found to be similar [3], the delayed maturation found in SOD1^G93A^ SCs may be specific to the premotor network.

5-HT has been described to be able to modulate chloride homeostasis by acting on NKCC1 in the maturing spinal locomotor network of the zebrafish [23]. In postnatal (P5–P7) rat MNs, 5-HT modulates chloride homeostasis specifically through KCC2; 5-HT increases the cell membrane expression of KCC2 [22,24]. Here, our pharmacological results in mouse prenatal MNs indicate that 5-HT likely modulates chloride homeostasis in mouse prenatal E17.5 MNs by acting on KCC2. This is in agreement with our previous data showing that the E_GABAAR_ relies on KCC2 and NKCC1 at early prenatal stages and KCC2 at later prenatal stages [17]. In a recent study, we have showed that KCC2 is reduced in E17.5 SOD1^G93A^ lumbar spinal MNs compared to the E17.5 lumbar spinal MNs of WT littermates [21]. Our data did not reveal any difference in the 5-HT-induced modulation of the E_GABAAR_ in SOD1^G93A^ and WT MNs, i.e., similar E_GABAAR_ hyperpolarization, even though we could have assumed that there was a reduced effect in SOD1^G93A^ MNs with diminished cell membrane expression of KCC2 [21]. This lack of difference between SOD1^G93A^ and WT MNs may be explained by the exogenous bath application of 5-HT in our protocol, which likely saturated the KCC2-dependent modulation of chloride homeostasis.

Our study using methysergide and ketanserin confirms that the effects of exogenous 5-HT on inward currents are related to 5-HTRs. The involvement of 5-HT_2A_R in mediating the locomotor function of 5-HT achieved by direct action on MNs is likely [25,26,27]. Additionally, 5-HT_2B_R and 5-HT_2C_R are also known to participate in locomotor function [28,29], although their participation in early developmental stages is less known. Similarly, spinal 5-HT_1A/1B/1D_R and 5-HT_7_R have a role in the control of locomotor activity, although they are not necessarily associated with direct action on MNs [26,27,30]. The cocktail methysergide plus ketanserin ensures the blockade of 5-HT_2_R, which is the preferential targets of these drugs, as well as 5-HT_1_R and 5-HT_7_R [31,32]. This allows us to disregard the catecholaminergic effects triggered by exogenous 5-HT [33] and highlights a 5-HTR-mediated effect. Interestingly, the 5-HTR-induced inward current was similar in SOD1^G93A^ and WT MNs, leading to the conclusion that SOD1^G93A^ and WT MNs are enriched with the same amount of 5-HTR. Additional pharmacological and anatomical data are needed to precisely determine the 5-HTR subtype(s) involved in the early hyperpolarization effect of 5-HT on MNs.

The involvement of 5-HT in ALS disease progression has been questioned [34,35]. The degeneration of 5-HT neurons has been described in ALS patients and in SOD1 (G86R) mice, and this degeneration has been correlated with spasticity symptoms [36]. Similarly, the degeneration of 5-HT neurons is necessary to trigger spasticity through the 5-HT_2B/C_ receptor in SOD1 (G37R) mice [37]. Interestingly, a brief period of antidepressant treatment during a critical time window (P5–P11, i.e., the transition from the neonatal to the juvenile state) can be detrimental in ALS mouse models [4], suggesting that the extracellular levels of 5-HT must be tightly regulated. All these data support the role of descending 5-HT inputs in the early deregulation of chloride homeostasis in E17.5 spinal lumbar MNs of the well-known SOD1^G93A^ ALS mouse model.

## 4. Materials and Methods

### 4.1. Ethical Considerations and Mouse Model

All procedures were carried out in accordance with the local ethics committee of the University of Bordeaux (authorization #5022, approved on 9 June 2017) and European Committee Council directives. All efforts were made to minimize animal suffering and reduce the number of animals used. B6SJLTgN(SOD1-G93A)/1Gur/J mice expressing the human G93A Cu/Zn superoxide dismutase (SOD1) mutation (Gly93→Ala substitution) were obtained from the Jackson Laboratory (Bar Harbor, ME, USA) (http://jaxmice.jax.org/strain/002726.html). Heterozygous B6SJL-TgN(SOD1-G93A)/1Gur/J mice (referred to as SOD1^G93A^ mice in the present report) were maintained by crossing heterozygous transgene-positive male mice with B6SJL F1 hybrid females (Janvier labs, Saint-Berthevin, France). The gestational period of SOD1^G93A^ mice was ~18.5 days, and embryonic day 0.5 (E0.5) was defined as the day after mating. Experiments were performed on E17.5 foetuses, i.e., foetuses collected 1 day before birth.

### 4.2. Dissection and Isolation of the ex vivo Embryonic Spinal Cord

Pregnant mice were sacrificed by cervical dislocation. A laparotomy was performed, and the foetuses were removed after cutting the uterine muscle. Foetuses of either sex were removed from their individual embryonic sacs and transferred to cooled artificial cerebrospinal fluid (aCSF) oxygenated with a 95% O_2_ and 5% CO_2_ mixture. The composition of the aCSF was (in mM): 114.5 NaCl, 3 KCl, 2 CaCl_2_-2H_2_O, 1 MgCl_2_-6H_2_O, 25 NaHCO_3_, 1 NaH_2_PO_4_-H_2_O, and 25 D-glucose, pH 7.4, osmolarity 307 mosmol/kg H_2_O). The selected foetuses were then decapitated, and their tails were preserved for subsequent genotyping. The SC was opened dorsally, and the meninges were removed. The preparations were placed in a recording chamber (ventral side up) and kept open under a nylon mesh. This exposed the MNs and made them accessible to the patch-clamp electrode (generally 1–2 MNs from each SC were recorded). The SC preparations were placed in a recording chamber and continuously superfused (~1.5 mL·min^−1^) with oxygenated aCSF. All experiments were carried out at constant temperature (30 °C). The experiments were performed blindly without knowledge of the genotype of the animals. The foetuses were genotyped at the genotyping facility of Neurocentre Magendie (Bordeaux Neurocampus). Genotyping was performed by standard PCR of mouse tail samples using established primers and a protocol provided by the Jackson Laboratory (Bar Harbor, ME, USA).

### 4.3. Immunohistochemistry

Immunohistochemistry was performed on coronal sections of lumbar SCs prepared as follows. Lumbar SC samples from E17.5 SOD1^G93A^ and WT embryos were fixed in 4% paraformaldehyde (PFA) for 2 h at room temperature. They were rinsed three times with 0.1 M phosphate-buffered saline (PBS) and then cryoprotected in 15% sucrose for 24 h followed by 30% sucrose for another 24 h. After they were placed in Tissue-Tek^®^ (O.C.T. Compound, Sakura^®^ Finetek) and frozen, the samples were sliced at the cervical 5–6 (C5–C6) and lumbar 4–5 (L4–L5) levels using a Leica 3050 S Cryostat. The sections (20 µm thick) were affixed to gelatinized slides and preserved at −25 °C until use. Each slide was rinsed three times with 0.1 M PBS, blocked with solution containing 2% bovine serum albumin (BSA), and then incubated for 48 h with primary antibodies prepared in PBST (1% Triton and 0.2% BSA). We processed the sections with an anti-serotonin (5-HT) antibody (1:5000, rabbit polyclonal, gift from Pr. G. Tramu, Univ. Bordeaux 1, for specificity see [38]). After three rinses, the slides were incubated for 2 h at room temperature with an Alexa Fluor^®^ 488-conjugated goat anti-rabbit IgG (H+L) secondary antibody (A-11008) (1:500, Fisher Scientific, Thermo Fisher Scientific, Illkirch, France) and then rinsed with 0.1 M PBS. After rinsing, the slides were mounted with an anti-fade reagent (Fluoromount, Electron Microscopy Sciences, Hatfield, PA, USA) and stored at 4 °C in darkness until observation under a confocal microscope. For better clarity, inverted images are presented.

### 4.4. Confocal Microscopy

Samples were visualized in the laboratory with a BX51 Olympus Fluoview 500 confocal microscope. Serial optical sections (0.2 μm thick) were obtained using a 60× oil-immersion objective. Lasers were selected according to the wavelength required for visualization. The global 5-HT staining density, which was used as an approximate estimation of fibre labelling, was assessed using ImageJ (Wayne Rasband, NIH, https://imagej.nih.gov/ij/).

### 4.5. Electrophysiological Procedures and Data Analysis

Patch-clamp electrodes were constructed from thin-walled single filamented borosilicate glass (1.5 mm outer diameter, Harvard Apparatus, Les Ulis, France) using a two-stage vertical microelectrode puller (PP-830, Narishige Scientific Instrument Lab., Setagaya-ku, Tokyo, Japan). The patch electrode resistances ranged from 3 to 5 MΩ. All recordings were made with an Axon Multiclamp 700B amplifier (Molecular Devices, Sunnyvale, CA, USA). The data were low-pass filtered (2 kHz) and acquired at 20 kHz on a computer via an analog-to-digital converter (Digidata 1322A, Molecular Devices) and data acquisition software (Clampex 10.3, Molecular Devices).

Motorized micromanipulators (Luigs & Neumann, Ratingen, Germany) were used to position the patch-clamp electrode on a visually identified lumbar 4–5 (L4–L5) MN using a CCD video camera (Zeiss Axiocam MR). The recorded MNs were located in the lateral column [21] and were identified by their pear-shaped cell bodies. Rin and Cm values also further confirmed the motoneuron (MN) identity of the recorded neurons [3].

For E_GABAAR_ assessment, gramicidin perforated patch-clamp recordings were conducted. For perforation, a gramicidin stock solution (Millipore Sigma) was dissolved in DMSO (Millipore Sigma) at a concentration of 10 μg·ml^−1^ and diluted to a final concentration of 10–20 μg·ml^−1^ in intracellular medium composed of (in mM): 130 KCl, 10 HEPES, 10 EGTA, and 2 MgATP, 284 mosmol/kg H_2_O, adjusted to pH 7.4 using 1 M KOH. The gramicidin stock solution and dilutions were prepared <1 h before each experiment. Pipette tips were filled with ~3 μL of filtered intracellular medium and subsequently back-filled with gramicidin solution. The GABA_A_R-specific agonist isoguvacine (50 µM, Bio-Techne, France) was pressure ejected (~3 psi; 50 ms) via a puff pipette placed in the vicinity of the motoneuron somata with a PicoSpritzer II (Parker Hannifin Corporation, Fairfield, NJ, USA) driven by a programmable Master 8 Stimulator/Pulse Generator (Master-8, A.M.P.I. Jerusalem, Israel). Isoguvacine application was repeated at different membrane voltages, allowing the assessment of the reversal potential of evoked GABA_A_R-related currents. The E_GABAAR_ was considered the E_Cl_ value because Cl^−^ ions are mainly involved in GABA_A_R currents in foetal MNs [39,40]). Measurements were corrected for liquid junction potentials (3.3 mV) calculated with the Clampex junction potential calculator. Using the Nernst equation, E_GABAAR_ was calculated to be 1.5 mV. Thus, to confirm that the perforated-patch configuration had not ruptured during the experiment, repetitive E_GABAAR_ tests were performed. Measurements that produced an E_GABAAR_ close to 0 mV were considered whole-cell recordings, and the cells were discarded. The Clampex membrane test was used to monitor the input resistance (Rin) and membrane capacitance (Cm); a −60 mV holding membrane potential and 5 mV steps (negative and positive, 40 ms duration) were chosen. The reversal potential of GABA_A_R (E_GABAAR_) was calculated by a linear fit (Prism 7 from GraphPad Software Inc., USA). Changes in the current amplitude were determined using Clampfit 10.3 (Molecular Devices).

### 4.6. Tissue Collection and Postmortem High-performance Liquid Chromatography (HPLC) Measurements

Lumbar samples from E17.5 SOD1^G93A^ and WT embryos were quickly removed, weighed, and placed in a tube at −80 °C for biochemical analysis. On the day of the analysis, tissue samples were homogenized using sonication in 0.1 N perchloric acid (HClO_4_) and centrifuged at 13,000 rpm for 30 min at 4 °C. The tissue content of the monoamine 5-HT was measured by a sensitive HPLC coupled to electrochemical detection (ECD) system according to previous protocols with some modifications [35,41]. Aliquots of the sample supernatants were injected into the HPLC column (Stability-C8, 150 mm × 4.6 mm, 5 μm; C.I.L.-Cluzeau) protected by a Brownlee–Newguard precolumn (RP-8, 15 × 3.2 mm, 7 μm; C.I.L.-Cluzeau) using a manual injector (7725i Rheodyne manual injector, C.I.L. Cluzeau, Sainte-Foy-La-Grande, France). The mobile phase was delivered at a flow rate of 1 mL/min using an HPLC pump (Beckman 128, system Gold, France) and was composed of the following (in mM): 70 NaH_2_PO_4_, 0.1 disodium EDTA and 2 octane sulfonic acid plus 7% methanol, adjusted to a pH of 4.2 with orthophosphoric acid and filtered through a 0.22 µm Millipore filter. 5-HT was detected using a coulometric cell (analytical cell 5011, ESA, France) coupled to a programmable detector (Coulochem II, ESA, Paris, France). The potential of the electrodes was set at +350 mV for oxidation and −270 mV for reduction. Output signals from the oxidation channel were acquired on a computer via an analog-to-digital converter (1401 Plus data acquisition unit, Cambridge Electronic Design, Cambridge, England) and data acquisition software (Spike 2 version 7.01, Cambridge Electronic Design, Cambridge, England). We measured the height of the signals corresponding to the time of elution of 5-HT and calculated the concentration in the sample using an appropriate calibration curve. Each day, standard solutions were injected before spinal cord samples were analysed to verify the HPLC system [41,42]. Under these conditions, the sensitivity for 5-HT was 8 pg/10 μL, and the signal/noise ratio was 3:1. The tissue content of 5-HT was expressed in pg/mg of tissue.

### 4.7. Pharmacology

In the gramicidin perforation experiments, GABA_A_R responses were isolated by using a cocktail of drugs containing 0.2 µM tetrodotoxin (TTX, Latoxan Laboratory, France), 4 mM kynurenic acid (Millipore Sigma), 10 µM (+)-tubocurarine (Millipore Sigma), 5 µM dihydro-β-erythroidine hydrobromide (DHΒE, Bio-Techne, Lille, France), and 3 µM strychnine (Millipore Sigma), which blocked voltage-dependent Na^+^ action potentials and glutamate, cholinergic, and glycinergic inputs to MNs, respectively. VU0240551 (Bio-Techne, France) (10 µM) was applied to specifically block KCC2. Serotonin (5-HT, 10 µM) was purchased from Millipore Sigma. Methysergide maleate (10 µM) and ketanserin tartrate (10 µM) (Bio-techne) were used to evaluate the involvement of 5-HTRs in the effect of 5-HT. Combined, these antagonists are known to efficiently block 5-HT_2A-C_R [31,32].

### 4.8. Statistical Analysis

GraphPad Prism 7 software was used to analyse all the data. The results are presented as the means ± the standard errors of the mean (SEMs). *n* is the number of MNs or SCs used in the analysis (always from at least three different litters). *p* < 0.05 (*), *p* < 0.01 (**), *p* < 0.001 (***) were considered significant. The significance of differences between two data sets was assessed by the Mann-Whitney test or Wilcoxon matched-pairs signed rank test for nonparametric data.

## Figures and Tables

**Figure 1 ijms-21-01107-f001:**
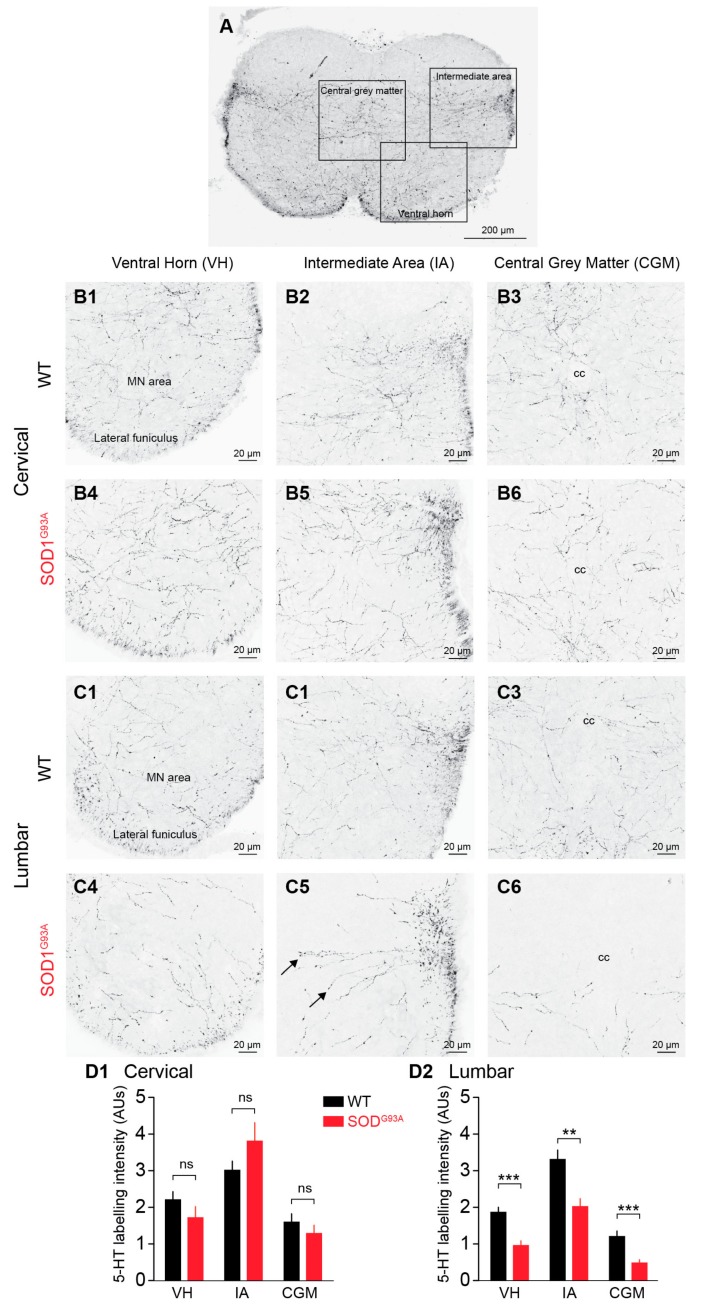
Descending 5-HT fibres in coronal sections of E17.5 WT and SOD1^G93A^ spinal cords (SCs). (**A**) Global 5-HT staining (dark labelling) in the three parts of the SC used for quantification (ventral horn, VH; intermediate area, IA; central grey matter, CGM). (**B1**–**B3**) 5-HT staining at the cervical level in WT mice. (**B4**–**B6**) 5-HT staining at the cervical level in SOD1^G93A^ mice. (**C1**–**C3**) 5-HT staining at the lumbar level in WT mice. (**C4**–**C6**) 5-HT staining at the lumbar level in SOD1^G93A^ mice. Note the 5-HT positive fibres projecting into the grey matter from the lateral funiculus (arrows) (**D1–D2**) Quantitative analysis of global 5-HT innervation in the VH, IA, and CGM at the cervical level (D1) and lumbar level (D2) in WT (black) and SOD1^G93A^ (red) SCs. ns, not significant; ** *p* < 0.01; *** *p* < 0.001, Mann-Whitney test); cc, central canal.

**Figure 2 ijms-21-01107-f002:**
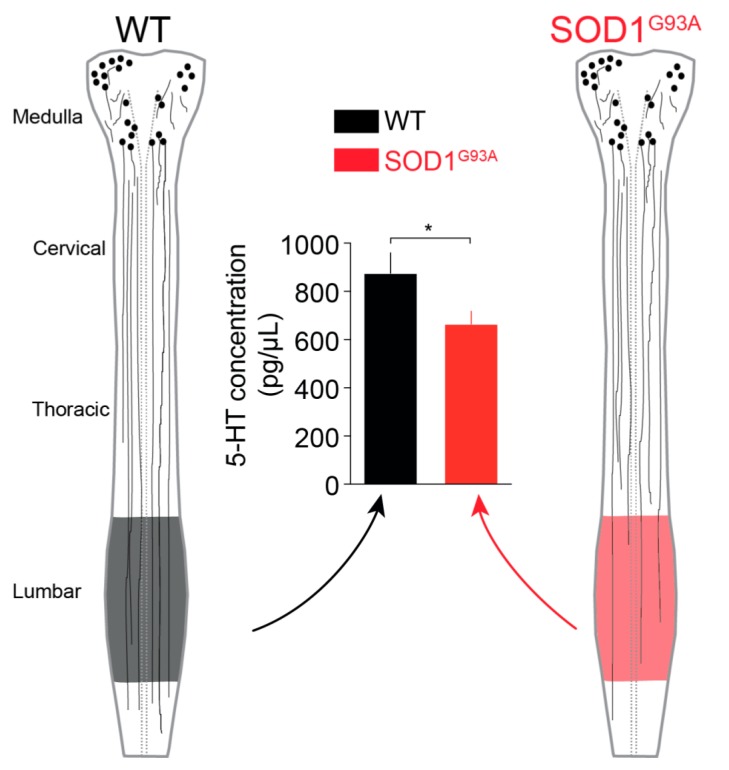
High-performance liquid chromatography (HPLC) assay of the monoamine 5-HT in E17.5 WT (black bar) and SOD1^G93A^ lumbar SCs (red bar). The schematic drawings on both sides of the graph depict the reduction in the density of descending 5-HT fibres from caudal raphe nuclei in the SC of SOD1^G93A^ mice compared to the SC of WT littermates. * *p* < 0.05, Mann-Whitney test.

**Figure 3 ijms-21-01107-f003:**
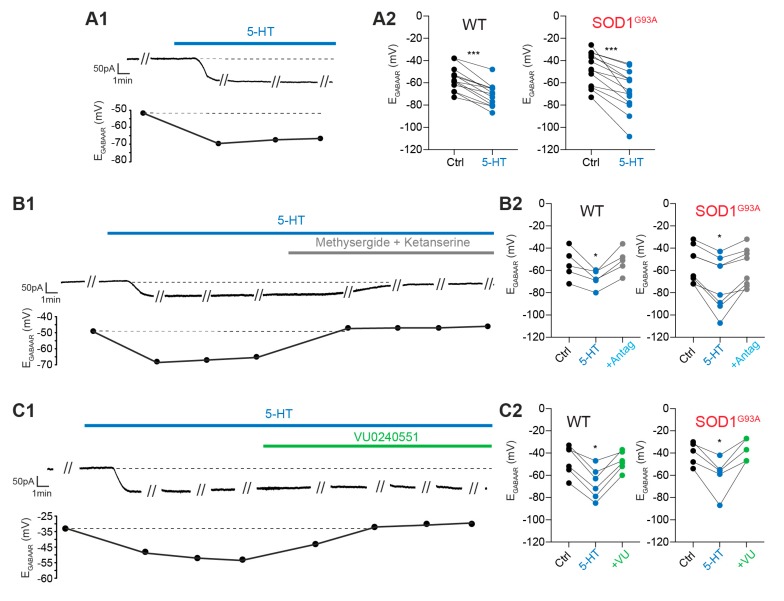
Modulation of the E_GABAAR_ by exogenous 5-HT application. (**A1**) 5-HT-induced inward current and E_GABAAR_ decrease in a representative E17.5 MN. (**A2**) E_GABAAR_ values measured in the same WT MNs (left panel) and SOD1^G93A^ MNs (right panel) under control conditions (black circles) and in the presence of 5-HT (blue circles). (**B1**) Involvement of the 5-HT_2_ receptor in the 5-HT-mediated modulation of the E_GABAAR_ in a representative E17.5 MN. (**B2**) Quantitative analysis of the E_GABAAR_ in the same WT MNs (left panel) and SOD1^G93A^ MNs (right panel) under control conditions (black circles), in the presence of 5-HT (blue circles) and in the presence of 5-HT + methysergide and ketanserine (grey circles). (**C1**) 5-HT hyperpolarized the E_GABAAR_ by acting on KCC2, as depicted in a representative E17.5 MN. (**C2**) Quantitative analysis of the E_GABAAR_ in the same WT MNs (left panel) and SOD1^G93A^ MNs (right panel) under control conditions (black circles), in the presence of 5-HT (blue circles) and in the presence of 5-HT + VU0240551 (green circles). Drugs were applied at 10 µM. * *p* < 0.05; *** *p* < 0.001, Wilcoxon matched-pairs signed rank test. Experiments in B–C are from the set of experiments used in A.

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
