# Peer review of "Implication of 5-HT in the Dysregulation of Chloride Homeostasis in Prenatal Spinal Motoneurons from the G93A Mouse Model of Amyotrophic Lateral Sclerosis"

_ijms, 2020, doi:10.3390/ijms21031107_

Round 1

Reviewer 1 Report

In the paper entitled “Implication of 5-HT in the dysregulation of the chloride homeostasis in prenatal spinal motoneurons from the G93A mouse model of Amyotrophic Lateral Sclerosis” Branchereau and colleagues address the implication of defective developmental 5-HT signaling in ALS pathogenesis.

ALS is a late-onset neurodegenerative disorder and is not commonly perceived as a neurodevelopmental disease. Nonetheless – as properly pointed out by the Authors – it is conceivable that early (developmental) abnormalities may concur to generate the pathologic phenotypes in adult individuals. This concept fully substantiates the experimental strategy.

Following a recent paper published by the same group, which showed altered chloride homeostasis – due to reduced KCC2 activity – in the lumbar spinal cord of embryos (E17.5) of the SOD1(G93A) ALS mouse model, the Authors here demonstrate that ALS mouse embryos have reduced descending 5-HT fibers and 5-HT content in the lumbar spinal cord tract compared to WT littermates. They also show – by patch-clamp techniques – that exogenous application of 5-HT to isolated, dorsally opened, spinal cords (lumbar tract) hyperpolarizes the reversal potential for the GABA-A receptors in motor neurons, and demonstrate – by pharmacological approaches – that 5-HT-2R receptor activity and upregulation of the KCC2 transporter are involved in such an effect.

Although in my opinion the presented histological and electrophysiological data are far to be causally correlated with each other in the context of ALS pathogenesis, the paper is scientifically sound and worth publication in IJMS provided that Authors convincingly reply to the following concern.

If I understood correctly, the Authors claim – based on previous work and data from Fig. 3A2 – that the reversal potential for the GABA-A receptors is more depolarized in SOD1(G93A) motor neurons compared to the WT counterpart under basal conditions (Ctrl). However, data in Fig. 3B2 and 3C2 seem not to be consistent with this statement. Authors should provide explanation for such a discrepancy, also in view that no statistical comparison between WT and SOD1(G93A) is shown.

MINOR CONCERN. There is some conceptual/grammar error here and there in the text (e.g., “SOD” and “WT” seem to be inverted in line 137; check for grammar in lines 223-224). Please check for this throughout the manuscript.

Author Response

In the paper entitled “Implication of 5-HT in the dysregulation of the chloride homeostasis in prenatal spinal motoneurons from the G93A mouse model of Amyotrophic Lateral Sclerosis” Branchereau and colleagues address the implication of defective developmental 5-HT signaling in ALS pathogenesis.

ALS is a late-onset neurodegenerative disorder and is not commonly perceived as a neurodevelopmental disease. Nonetheless – as properly pointed out by the Authors – it is conceivable that early (developmental) abnormalities may concur to generate the pathologic phenotypes in adult individuals. This concept fully substantiates the experimental strategy.

Following a recent paper published by the same group, which showed altered chloride homeostasis – due to reduced KCC2 activity – in the lumbar spinal cord of embryos (E17.5) of the SOD1(G93A) ALS mouse model, the Authors here demonstrate that ALS mouse embryos have reduced descending 5-HT fibers and 5-HT content in the lumbar spinal cord tract compared to WT littermates. They also show – by patch-clamp techniques – that exogenous application of 5-HT to isolated, dorsally opened, spinal cords (lumbar tract) hyperpolarizes the reversal potential for the GABA-A receptors in motor neurons, and demonstrate – by pharmacological approaches – that 5-HT-2R receptor activity and upregulation of the KCC2 transporter are involved in such an effect.

Although in my opinion the presented histological and electrophysiological data are far to be causally correlated with each other in the context of ALS pathogenesis, the paper is scientifically sound and worth publication in IJMS provided that Authors convincingly reply to the following concern.

If I understood correctly, the Authors claim – based on previous work and data from Fig. 3A2 – that the reversal potential for the GABA-A receptors is more depolarized in SOD1(G93A) motor neurons compared to the WT counterpart under basal conditions (Ctrl). However, data in Fig. 3B2 and 3C2 seem not to be consistent with this statement. Authors should provide explanation for such a discrepancy, also in view that no statistical comparison between WT and SOD1(G93A) is shown.

We understand reviewer #1’s concern. We have added the statistical analysis between SOD1G93A and WT EGABAAR values showed in A2. Data in B2 and C2 are from the same series of experiment used in A2, sometimes after 5-HT washout. Therefore values may have fluctuated. We did not detect any significant difference between SOD1G93A and WT in B2 and C2, likely because of the variability of EGABAAR before and after 5-HT application and because of the insufficient amount of data in B2 and C2.

MINOR CONCERN. There is some conceptual/grammar error here and there in the text (e.g., “SOD” and “WT” seem to be inverted in line 137; check for grammar in lines 223-224). Please check for this throughout the manuscript.

We are sorry for those errors. The manuscript has been corrected by AJE.

Reviewer 2 Report

Martin E. et al., described the implication of 5-HT in SOD1-G93A-expression mice using immunostaining, HPLC, and electrophysiology. First, this reviewer would like to ask the authors why they use wild type mice not wildtype SOD1-expressing one as a control. I believe it would be important. Second, I think ‘SOD’ abbreviation seems to be inappropriate because it is confusing whether SOD1 WT-expressing system. ‘mutSOD’ or ‘SOD-G93A’ seems to be appropriate and it would be easily read.

Author Response

Martin E. et al., described the implication of 5-HT in SOD1-G93A-expression mice using immunostaining, HPLC, and electrophysiology. First, this reviewer would like to ask the authors why they use wild type mice not wildtype SOD1-expressing one as a control. I believe it would be important.

Experiments have been performed on SOD1G93A mice and WT littermates. Experiments using SOD1G93A and SODWT plus WT littermates as control did not highlight differences between SODWT plus WT littermates controls (e.g., Pieri et al. Exp. Neurol. 2013). We believe that using SODWT and WT littermates is the most correct but we keep convinced that our control is appropriate.

Second, I think ‘SOD’ abbreviation seems to be inappropriate because it is confusing whether SOD1 WT-expressing system. ‘mutSOD’ or ‘SOD-G93A’ seems to be appropriate and it would be easily read.

We have replaced SOD1 by SOD1G93A.

Reviewer 3 Report

This manuscript investigated the Implication of 5-HT in the dysregulation of the chloride homeostasis in prenatal spinal motoneurons from the G93A mouse model of Amyotrophic Lateral Sclerosis.

I have no comments concerning the work, it is detailed, well written and the conclusions are sounding.

The following minor issues need to be addressed:

line 24: reregulated?

line 39, reference [2]: Benedetti and co-workers (2016) have clearly shown that the SOD1 G93R mutation is associated to developmental motor neurons defects, spinal neurons hyperexcitability and behavioral alterations in zebrafish embryos and larvae, therefore this publication deserves to be included in the references here for a more complete Introduction (Benedetti, L., Ghilardi, A., Rottoli, E., De Maglie, M., Prosperi, L., Perego, C., Baruscotti, M., Bucchi, A., Del Giacco, L., Francolini, M. (2016) INaP selective inhibition reverts precocious inter- and motorneurons hyperexcitability in the Sod1-G93R zebrafish ALS model. Scientific Reports. 2016 Apr 15;6:24515. doi: 10.1038/srep24515.).

line 54: ‘where’ instead of ‘were’.

line 203: what does “down-regulated” mean? Please explain

line 223-224: The sentence needs to be checked.

line 235: Check the journal guidelines to see whether it is mandatory to report here the number of authorization for animal use.

Abbreviations: introduce 5-HTRs > 5-HT receptors

Author Response

This manuscript investigated the Implication of 5-HT in the dysregulation of the chloride homeostasis in prenatal spinal motoneurons from the G93A mouse model of Amyotrophic Lateral Sclerosis.

I have no comments concerning the work, it is detailed, well written and the conclusions are sounding.

The following minor issues need to be addressed:

line 24: reregulated?

Read deregulated. It has been corrected.

line 39, reference [2]: Benedetti and co-workers (2016) have clearly shown that the SOD1 G93R mutation is associated to developmental motor neurons defects, spinal neurons hyperexcitability and behavioral alterations in zebrafish embryos and larvae, therefore this publication deserves to be included in the references here for a more complete Introduction (Benedetti, L., Ghilardi, A., Rottoli, E., De Maglie, M., Prosperi, L., Perego, C., Baruscotti, M., Bucchi, A., Del Giacco, L., Francolini, M. (2016) INaP selective inhibition reverts precocious inter- and motorneurons hyperexcitability in the Sod1-G93R zebrafish ALS model. Scientific Reports. 2016 Apr 15;6:24515. doi: 10.1038/srep24515.).

This interesting reference has been included. We apologize for having omitted this reference in our first draft.

line 54: ‘where’ instead of ‘were’.

Corrected

line 203: what does “down-regulated” mean? Please explain

Down-regulated has been replaced by reduced.

line 223-224: The sentence needs to be checked.

This sentence has been rewritten.

line 235: Check the journal guidelines to see whether it is mandatory to report here the number of authorization for animal use.

The authorization # has been included.

Abbreviations: introduce 5-HTRs > 5-HT receptors

Done.

Round 2

Reviewer 2 Report

I satisfied.